# Enhanced Fluctuations in Facial Pore Size, Redness, and TEWL Caused by Mask Usage Are Normalized by the Application of a Moisturizer

**DOI:** 10.3390/jcm11082121

**Published:** 2022-04-11

**Authors:** Kukizo Miyamoto, Yoko Munakata, Xianghong Yan, Gaku Tsuji, Masutaka Furue

**Affiliations:** 1Research and Development, Kobe Innovation Center, Procter and Gamble Innovation GK, Kobe 651-0088, Japan; munakata.y@pg.com (Y.M.); yan.xh@pg.com (X.Y.); 2Research and Clinical Center for Yusho and Dioxin, Kyushu University Hospital, Fukuoka 812-8582, Japan; tsuji.gaku.893@m.kyushu-u.ac.jp; 3Department of Dermatology, Graduate School of Medical Sciences, Kyushu University, Fukuoka 812-8582, Japan; furuemasutaka00@yahoo.co.jp

**Keywords:** facial imaging, COVID-19, mask usage, skin hydration, TEWL, pore size, redness, fluctuation, *Galactomyces* ferment filtrate-containing skin care formula, GFF Pitera™

## Abstract

Mask wearing is described as one of the main public health measures against COVID-19. Mask wearing induces various types of subjective and objective facial skin damage, such as hair pore dilatation and redness. Facial pore size and redness show morning-to-evening intra-day fluctuations. It remains unknown whether mask usage affects fluctuations in pore size and redness. We measured facial skin hydration, transepidermal water loss (TEWL), pore size, and redness four times a day for 6 weeks in 20 healthy young women. After a 2-week no-mask-usage period (baseline period), all subjects wore unwoven masks for 2 weeks; then, for the following 2 weeks, they applied masks after the topical application of a moisturizer containing a *Galactomyces* ferment filtrate (GFF) skin care formula (Pitera™). We demonstrated that mask wearing significantly increased the intra-day fluctuations of pore size, redness, and TEWL. In addition, significant correlations were evident among these three parameters. Notably, these mask-induced skin changes were significantly improved, achieving a return to baseline levels, by the application of a GFF-containing moisturizer. In conclusion, mask wearing aggravates intra-day fluctuations in pore size and redness. Appropriate moisturization can minimize this mask-related skin damage, most likely by normalizing the elevated TEWL.

## 1. Introduction

Mask wearing is described as one of the main public health measures to prevent the spread of COVID-19, and masks have quickly become ubiquitous during the COVID-19 pandemic [1]. Although mask wearing is useful for reducing the risk of viral infection and transmission, it frequently induces various subjective (pruritus, dryness, pricking, and pain) and objective (acne, redness, indentation, aggravation of preexisting skin diseases, and pigmentation) facial skin troubles [2,3]. Among these types of mask-related skin damage, acne is one of the most important dermatological adverse events, especially for women [4,5]. Mask-induced acne and its aggravation are now widely called maskne [6,7,8]. 

Acne is a condition involving the chronic inflammation of hair follicles [9]. The plugging of hair pores with sebum and cellular debris, bacterial colonization by *Cutibacterium acnes*, and subsequent inflammatory processes are involved in the pathomechanisms of acne [9]. It has been proven that mask wearing damages the skin barrier and increases transepidermal water loss (TEWL) in facial skin [10,11,12]. It also accelerates facial redness and induces hair pore dilatation and sebum hyperproduction [10,11,12,13]. Mask-related pore dilatation and redness are potentially involved in the pathogenesis of maskne. It is known that facial skin variables, such as pore size and redness, show morning-to-evening intra-day fluctuations in daily life [14]. However, the effects of mask usage on the intra-day fluctuations of skin conditions and the best options for preventive measures remain unknown.

In this study, we measured facial skin hydration, TEWL, pore size, and redness four times a day for 6 weeks in 20 healthy young women. After a 2-week no-mask-usage period (baseline period), all subjects wore unwoven masks for 2 weeks; then, in the following 2 weeks, they applied a mask after the topical application of a moisturizer containing a *Galactomyces* ferment filtrate (GFF) skin care formula (Pitera™). We found that hair pores were significantly enlarged during the mask-wearing period compared with the baseline period. Moreover, mask wearing significantly increased the fluctuation width (Δ fluctuation) of pore size, TEWL, and redness during the mask-wearing period. In addition, the GFF-containing moisturizer normalized these mask-induced skin changes. 

## 2. Materials and Methods

### 2.1. Subjects and Study Protocol

A skin assessment following mask usage was conducted in 20 healthy Japanese women aged 22–45 years (mean ± SD, 33.4 ± 3.5). The clinical investigation was conducted over 6 weeks and divided into three 2-week phases as follows: phase I period, baseline without mask wearing; phase II period, mask wearing for more than 6 h per day; and phase III period, mask wearing with the topical application of a GFF-containing skin care formula (Pitera, 1.3 mL twice daily). In the phase III treatment period, all 20 subjects were allowed to use the GFF-containing skin care formula (Pitera) for their facial care.

The study was conducted from 10 February 2021 to 30 April 2021 in Osaka, Japan. The study protocol and trial method were reviewed and approved by the IRB based on GCP compliance, and the subjects signed an informed consent form prior to participation. 

### 2.2. Measurement of Pore Size, Redness, Skin Hydration, and TEWL

Facial skin images were captured by the eMR Pro portable self-facial imaging system developed by P&G Company (Kobe, Japan, IP filing ref. number AA1280), as previously reported [14]. Facial pore size and skin redness were analyzed using eMR calculation software (P&G) [14]. Skin hydration and TEWL were measured using GPSkin Barrier (GPOWER Inc., Seoul, Korea). The skin condition measurements were performed on the cheek skin four times a day: in the morning after waking up, in the morning after face washing, in the afternoon between 14–17 p.m., and in the evening after face washing. Values are expressed as arbitrary units (AU).

### 2.3. Statistical Analysis

SPSS 22 (IBM^®^ SPSS^®^ Statistics, New York, NY, USA) for Windows 2010 was used for statistical analyses. Data were analyzed by nonparametric, paired, and unpaired tests, including, if appropriate, an analysis of (co-)variance and Welch’s *t*-test. Pearson’s product–moment correlation coefficient analysis was performed for correlations among the intra-day ∆ fluctuation in TEWL, skin hydration, pore size, and redness. Significance was assumed at *p* values less than 0.05. 

## 3. Results

Given that we measured skin condition four times a day, we first compared the intra-day averages of skin hydration, TEWL, pore size, and redness in the baseline and mask-wearing periods. Mask usage tended to increase the measurement values for all four parameters; significant elevation was observed for pore size (Table 1) compared with levels in the baseline period. 

The measurement values for these four parameters showed morning-to-evening fluctuations [14]. In addition, intra-day fluctuations were noted in this study (Appendix A). A significant mask-induced enlargement of pore size was observed in the afternoon when the subjects wore their masks (Appendix A).

We then compared the intra-day Δ fluctuations of these parameters in the no-mask baseline period and the mask-wearing period (Appendix A). As shown in Table 2, the intra-day Δ fluctuations of TEWL, pore size, and redness were significantly higher in the mask-wearing period than in the baseline period. 

Notably, the Δ fluctuation in TEWL was significantly correlated with those of pore size and redness (Figure 1). These results suggest that skin barrier impairment was potentially associated with pore enlargement and increased redness.

In addition, a significant correlation was evident between the Δ fluctuation in pore size and that of redness (Appendix A).

As mask wearing aggravated the skin condition parameters, we next investigated whether the topical application of a moisturizer improved or normalized these abnormalities. As shown in Appendix A, a topical treatment with a GFF-containing moisturizer was effective at increasing the intra-day average skin hydration level and decreasing the intra-day average TEWL. In parallel, moisturizer application significantly reduced the mask-induced elevation of the intra-day average pore size, returning it to the baseline level (Figure 2). 

Moreover, the mask-related aggravation of the intra-day Δ fluctuations in pore size, redness, and TEWL was significantly improved by the topical application of a GFF-containing moisturizer (Figure 3, Figure 4 and Figure 5). The intra-day Δ fluctuation in skin hydration was also stabilized at a low level (Figure 3).

## 4. Discussion

Previous studies demonstrated that mask usage aggravates TEWL [10,11,12]. Facial redness and hair pore size are also increased by mask wearing [10,11,12,13]. However, the measurement values for these parameters are not stable; they show significant intra-day fluctuation [14,15]. Therefore, once-daily measurement may not be sufficient to capture the extent of mask-induced effects on skin conditions.

In this study, we measured skin hydration, TEWL, pore size, and redness four times a day for 6 weeks, including three phases: a no-mask baseline (2 weeks), a mask-wearing period (2 weeks), and a mask-wearing period with moisturizer treatment (2 weeks). We confirmed that mask usage significantly enlarged the intra-day average of hair pore size. In addition, it enhanced the intra-day Δ fluctuations in pore size, redness, and TEWL. 

The pathomechanisms of the mask-induced alterations of these parameters remain unknown. However, possible suspected explanations are as follows. First, the humidity and temperature of the skin surface are elevated by mask usage [10]; hence, the humid and warm atmosphere of the skin surface rapidly decreases after removing the mask. This rapid change in skin surface conditions may cause epidermal corneal damage, resulting in unstable and irritated skin. Second, the facial mask is in close contact with the skin surface. Following facial movement, fine friction or scratching may continuously occur on the skin surface. It is known that mechanical scratching stimulates epidermal keratinocytes to release large amounts of cyto/chemokines, which chemoattract inflammatory cells such as neutrophils [16,17,18]. Therefore, physicomechanical stress is likely involved in mask-induced skin irritation. Such a notion has been described in the literature as the mask-related Koebner phenomenon in maskne and psoriasis [19,20].

It should be stressed that the aforementioned assumption was underpinned, at least in part, by the fact that skin barrier impairment (elevated Δ fluctuation in TEWL) was significantly associated with increased Δ fluctuations in pore size and redness. Therefore, we further investigated whether a topical moisturizer that upregulated skin hydration and downregulated TEWL could normalize the elevated Δ fluctuations in skin parameters. The GFF-containing moisturizer indeed improved the mask-induced aggravation of the intra-day Δ fluctuations in pore size and redness as well as TEWL. The present results well coincided with a previous report demonstrating that a GFF-containing moisturizer minimized intra-day Δ fluctuations in pore size and redness in healthy subjects without mask usage [14]. In addition to its high moisturizing ability, as shown in Appendix A, GFF is known to increase hyaluronic acid production [21], decrease sebum secretion [14], increase skin barrier-related proteins [22], and provide anti-oxidative effects [23]. These functional properties of GFF may potentially contribute to reducing mask-induced skin damage.

There are several limitations in this study, including its small sample size (20 women) and the fact that it was a short-term (6 weeks), unblinded clinical investigation. This trial was a one-arm, longitudinal study. All subjects used the GFF-containing Pitera in the treatment phase. As we did not enroll subjects treated with conventional emollients, we could not determine whether the stabilization of pore size and redness is secondary to GFF itself or rather simply due to decreased TEWL by moisturization. It also remains unclear whether GFF can stabilize the mechanical irritation of epidermal keratinocytes.

In conclusion, mask usage did enhance intra-day Δ fluctuations in pore size, redness, and TEWL. These mask-related skin changes can be stabilized by the topical application of a GFF-containing skin care formula. Further studies are warranted to clarify whether these mask-related skin changes underscore mask-induced Koebner dermatosis.

## Figures and Tables

**Figure 1 jcm-11-02121-f001:**
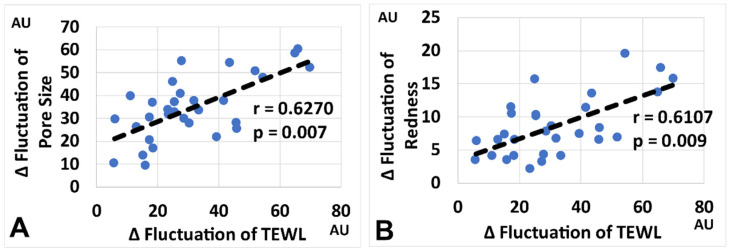
(**A**). Correlation between Δ fluctuation in TEWL and Δ fluctuation in pore size. (**B**). Correlation between Δ fluctuation in TEWL and Δ fluctuation in redness.

**Figure 2 jcm-11-02121-f002:**
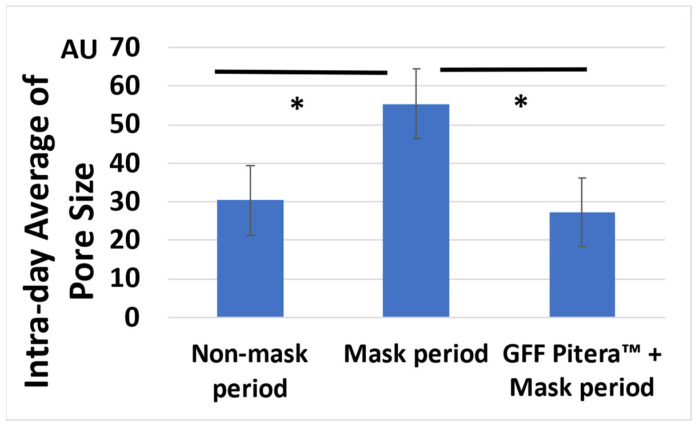
Effect of moisturization on the mask-induced enlargement of pore size. *; *p* < 0.05.

**Figure 3 jcm-11-02121-f003:**
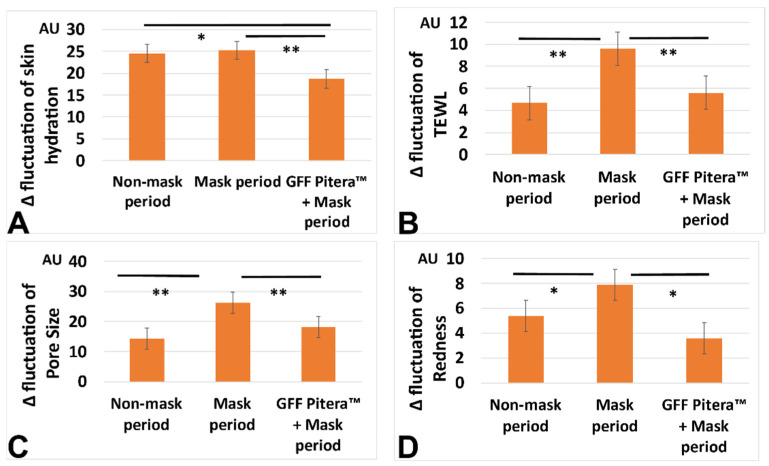
Effects of moisturization on the Δ fluctuation in skin hydration (**A**), Δ fluctuation in TEWL (**B**), Δ fluctuation in pore size (**C**), and Δ fluctuation in redness (**D**). *; *p* < 0.05. **; *p* < 0.01.

**Figure 4 jcm-11-02121-f004:**
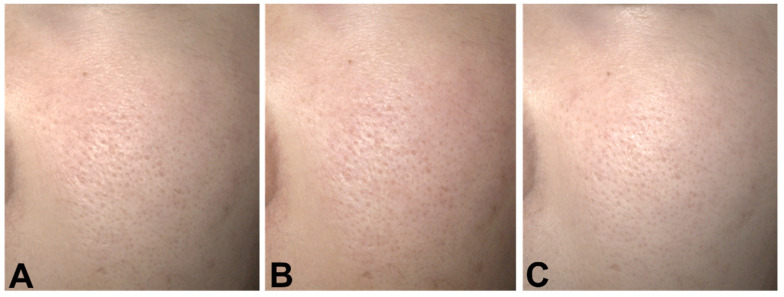
Case 1. Facial images taken after face washing in the morning. (**A**) Final day of non-mask period (phase I), (**B**) Final day of mask period (phase II), (**C**) Final day of GFF Pitera™ + mask period (phase III).

**Figure 5 jcm-11-02121-f005:**
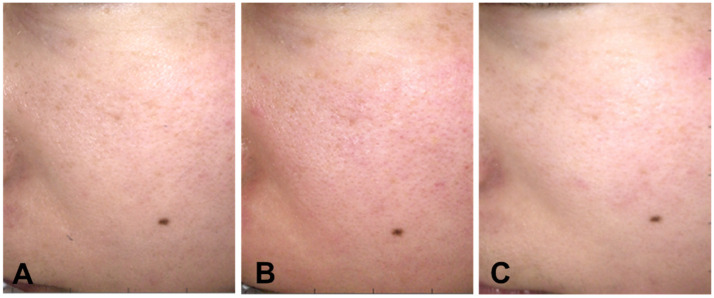
Case 2. Facial images taken after face washing in the morning. (**A**) Final day of non-mask period (phase I), (**B**) Final day of mask period (phase II), (**C**) Final day of GFF Pitera™ + mask period (phase III).

**Table 1 jcm-11-02121-t001:** Intra-day averages of facial skin variables.

Intra-Day Average	No-Mask Baseline Period	Mask-Wearing Period	*p* Value
Skin hydration (AU)	45.51 ± 3.2053	45.91 ± 4.1893	0.342
TEWL (AU)	44.93 ± 2.9413	45.62 ± 3.8747	0.285
Pore size (AU)	30.33 ± 4.1542	55.44 ± 7.8345 *	0.015
Redness (AU)	63.26 ± 3.6485	63.37 ± 4.3198	0.687

AU, arbitrary units. *, statistically significant.

**Table 2 jcm-11-02121-t002:** Intra-day Δ fluctuation in facial skin variables.

Intra-Day Δ Fluctuation	No-Mask Baseline Period	Mask-Wearing Period	*p* Value
Skin hydration (AU)	24.53 ± 3.5821	25.23 ± 4.042	0.288
TEWL (AU)	4.67 ± 1.8295	9.63 ± 3.5148 *	0.005
Pore size (AU)	14.34 ± 3.3655	26.24 ± 5.9189 *	0.003
Redness (AU)	5.41 ± 2.6613	7.88 ± 3.8652 *	0.026

AU, arbitrary units. *, statistically significant.

## Data Availability

The data presented in this study are available on request from the corresponding author. The data are not publicly available because of privacy restrictions.

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
