# Peer review of "Enhanced Fluctuations in Facial Pore Size, Redness, and TEWL Caused by Mask Usage Are Normalized by the Application of a Moisturizer"

_jcm, 2022, doi:10.3390/jcm11082121_

Round 1

Reviewer 1 Report

Minor comment;

Face mask is described as " the main preventive health measure against covid-19" thereby omiting the importance of social distancing. I would therefore amend to "a preventive health measure" (or an important health measure). In the introduction mask wearing is described as one of the main public health measures - which is more appropriate.

Major Comment

The study shows some interesting findings. the first one is that mask wearing results in an increased in pore size which is most liekly secondary to TEWL. The biggest challange of this manuscript is the lack of control arm - spefically patients applying a bland emollient. Can we confidently establish the stabilization of pore size is secondary to GFF or rather simply to decreased in TEWL from moisturization?

Author Response

Reply to the Reviewer 1

Minor comment;

Face mask is described as " the main preventive health measure against covid-19" thereby omiting the importance of social distancing. I would therefore amend to "a preventive health measure" (or an important health measure). In the introduction mask wearing is described as one of the main public health measures - which is more appropriate.

→ Thank you very much for your helpful comment. We agree with your comment. According to your comment, we amended the line 21 and line 44 as underlined.

“ Mask wearing is described as one of the main public health measures”

Major Comment

The study shows some interesting findings. the first one is that mask wearing results in an increased in pore size which is most liekly secondary to TEWL. The biggest challange of this manuscript is the lack of control arm - spefically patients applying a bland emollient. Can we confidently establish the stabilization of pore size is secondary to GFF or rather simply to decreased in TEWL from moisturization?

→ Thank you very much for your helpful comment. We agree with your comment. We did not have a control arm. Therefore, we added this point into the limitation of this study as follows.

Line 192 to 195

“As we did not enroll control healthy volunteers treated with conventional emollients, we could not determine whether the stabilization of pore size and redness is secondary to GFF itself or rather simply to decreased TEWL by moisturization.”

Thank you very much again for your valuable comments. We hope the revised article is now suitable for publication in JCM.

Reviewer 2 Report

This article is well connected to the current situation of the COVID-19 pandemic. Perhaps the authors could add a bit more background info to readers why they choose the Pitera in this study, not other traditional moisturizer such as urea cream or heparinoid. I wonder the difference between the Pitera and traditional moisturizer in stabilizing the mask-related skin changes. So it will be more convincing that authors can add revelant information.

Author Response

Reply to the Reviewer 2

This article is well connected to the current situation of the COVID-19 pandemic. Perhaps the authors could add a bit more background info to readers why they choose the Pitera in this study, not other traditional moisturizer such as urea cream or heparinoid. I wonder the difference between the Pitera and traditional moisturizer in stabilizing the mask-related skin changes. So it will be more convincing that authors can add revelant information.

→ Thank you very much for your helpful comment. We agree with your comment. We did not have a control arm. Therefore, we added this point into the limitation of this study as follows.

Line 192 to 195

“As we did not enroll control healthy volunteers treated with conventional emollients, we could not determine whether the stabilization of pore size and redness is secondary to GFF itself or rather simply to decreased TEWL by moisturization.”

Thank you very much again for your valuable comments. We hope the revised article is now suitable for publication in JCM.

Round 2

Reviewer 1 Report

Thank you for updating your manuscript. I would recommend comparing the treatment population to a control arm.

Author Response

Reply to the Reviewer 2 (re-revision)

Thank you for updating your manuscript. I would recommend comparing the treatment population to a control arm.

→ Thank you very much for your valuable comment. First, we have to apologize to confuse you about the trial protocol. This is one arm trial including phase I period, baseline without mask wearing; phase II period, mask wearing for more than 6 h per day; and phase III period, mask wearing with topical application of a GFF-containing skin care formula (Pitera, 1.3 ml twice daily). In the phase III treatment period, all 20 subjects were allowed to use the GFF-containing skin care formula (Pitera) for the facial care. We compared the results of phase I, II and III in Figure 2, Figure 3 and Supplementary Figure S4.Therefore, we added the following underlined sentences in line 78 to line 84.Line 78 to line 84“The clinical investigation was conducted over 6 weeks and divided in three 2-week phases as follows: phase I period, baseline without mask wearing; phase II period, mask wearing for more than 6 h per day; and phase III period, mask wearing with topical application of a GFF-containing skin care formula (Pitera, 1.3 ml twice daily). In the phase III treatment period, all 20 subjects were allowed to use the GFF-containing skin care formula (Pitera) for the facial care.” We also added the following underlined sentence in the limitation.Line 194 to 198“This trial was a one arm, longitudinal study. All subjects used the GFF-containing Pitera in the treatment phase. As we did not enroll subjects treated with conventional emollients, we could not determine whether the stabilization of pore size and redness is secondary to GFF itself or rather simply to decreased TEWL by moisturization.” Thank you very much again for your reviewing our revised article. We hope this re-revised article is now suitable for publication in JCM. 
